# CONSTRAINT-AWARE FEDERATED LEARNING: MULTI-RESOURCE OPTIMIZATION VIA DUAL ASCENT

## ABSTRACT

We present CAFL (Constraint-Aware Federated Learning), a principled approach for multi-resource optimization in federated learning that simultaneously manages energy, communication, memory, and thermal constraints through dual ascent. Unlike existing methods that optimize primarily for convergence, CAFL formulates federated learning as a constrained optimization problem and employs Lagrangian dual methods to dynamically adapt layer freezing, compression levels, and batch sizing. We provide theoretical convergence guarantees for our dual ascent controller and demonstrate that CAFL preserves training effectiveness through token-budget preservation while achieving significant resource savings. Experimental results on character-level language modeling demonstrate a 70.5% reduction in energy consumption, 95.3% lower communication cost, and 23% memory savings compared to FedAvg, while maintaining comparable convergence in training loss.

## 1 INTRODUCTION

Federated learning (FL) enables collaborative model training across distributed devices while preserving data privacy (McMahan et al., 2017). However, practical deployment on edge devices faces critical resource constraints including limited energy, communication bandwidth, memory capacity, and thermal budgets. Traditional FL approaches optimize primarily for convergence rate and accuracy, often ignoring these practical limitations that determine real-world feasibility.

Consider a federated learning scenario with $n$ clients, where each client $i$ has local dataset $\mathcal{D}_i$, local objective $F_i(w) = \frac{1}{|\mathcal{D}_i|} \sum_{(x,y) \in \mathcal{D}_i} \ell(f_w(x), y)$, and the global objective is $F(w) = \sum_{i=1}^{n} p_i F_i(w)$ where $p_i = \frac{|\mathcal{D}_i|}{\sum_j |\mathcal{D}_j|}$. Standard federated optimization algorithms like FedAvg solve:

$$\min_{w} F(w) = \sum_{i=1}^{n} p_i F_i(w) \tag{1}$$

However, this formulation neglects the fundamental resource constraints that govern practical deployment. Edge devices have finite energy budgets $E_{\text{budget}}$, communication limits $C_{\text{budget}}$, memory constraints $M_{\text{budget}}$, and thermal thresholds $T_{\text{budget}}$. Exceeding these constraints leads to device failures, user dissatisfaction, or system shutdowns.

We reformulate federated learning as a multi-constraint optimization problem:

$$\min_{w,\phi} \quad F(w) \tag{2}$$

$$\text{s.t.} \quad E(\phi) \leq E_{\text{budget}}, \quad C(\phi) \leq C_{\text{budget}}, \quad M(\phi) \leq M_{\text{budget}}, \quad T(\phi) \leq T_{\text{budget}} \tag{3}$$

where $\phi = (k, s, b, c)$ represents adaptive policy parameters: layer freezing depth $k$, local training steps $s$, batch size $b$, and compression level $c$. The resource consumption functions $E(\phi), C(\phi), M(\phi), T(\phi)$ model energy usage, communication overhead, memory pressure, and thermal generation respectively.

**Contributions.** Our work makes the following contributions:

- **Multi-resource FL formulation:** We formalize federated learning with multiple resource constraints as a constrained optimization problem with rigorous resource modeling.

- **Dual ascent controller:** We develop a theoretically grounded dual ascent algorithm for managing multiple constraints simultaneously with convergence guarantees.
- **Token-budget preservation:** We introduce a novel technique to maintain training effectiveness under resource constraints by preserving total processed tokens through gradient accumulation.
- **Comprehensive evaluation:** We demonstrate significant resource savings (70.5% energy, 95.3% communication) with acceptable accuracy trade-offs on language modeling tasks.

## 2 RELATED WORK

**Resource-constrained federated learning.** Prior work has addressed individual resource constraints in FL. Kang et al. (2017) focuses on energy-efficient inference partitioning. Wang et al. (2019) addresses communication efficiency through client selection. Li et al. (2020) introduces system heterogeneity awareness. However, none provide a unified framework for simultaneous multi-resource optimization.

**Communication-efficient federated learning.** Substantial research has targeted communication reduction through gradient compression (Lin et al., 2018), sparsification (Wangni et al., 2018), and quantization (Alistarh et al., 2017). Our work complements these by integrating compression into a broader multi-resource optimization framework.

**Dual methods in distributed optimization.** Dual ascent and ADMM have been applied to distributed ML (Boyd et al., 2011). Zhang et al. (2012) uses dual methods for communication-constrained optimization. We extend this to federated learning with multiple heterogeneous resource constraints.

## 3 METHOD

### 3.1 RESOURCE MODELING

We model four critical resource types:

**Energy Consumption.** Energy usage is proportional to computational workload:

$$E(\phi) = \alpha_E \cdot \mathcal{P}(k) \cdot s \cdot b \cdot \tau \tag{4}$$

where $\mathcal{P}(k)$ is the number of trainable parameters with $k$ unfrozen layers, $s$ is local steps, $b$ is batch size, $\tau$ is clients per round, and $\alpha_E$ is an energy scaling factor.

**Communication Cost.** Communication overhead depends on model size and compression:

$$C(\phi) = \beta_C \cdot \mathcal{P}(k) \cdot \rho(c) \cdot \tau \tag{5}$$

where $\rho(c)$ is the compression ratio (bytes per parameter) at compression level $c$, and $\beta_C$ accounts for protocol overhead.

**Memory Pressure.** Memory usage scales with batch size and model size:

$$M(\phi) = \gamma_M \cdot (\mathcal{P}(k) \cdot b + M_{\text{base}}) \tag{6}$$

where $M_{\text{base}}$ is baseline memory consumption and $\gamma_M$ is a scaling factor.

**Thermal Generation.** Thermal load depends on computational intensity:

$$T(\phi) = \delta_T \cdot (T_{\text{base}} + \epsilon_T \cdot s + \zeta_T \cdot b) \tag{7}$$

where $T_{\text{base}}$ is baseline temperature, and $\epsilon_T, \zeta_T$ model step and batch size thermal contributions.

### 3.2 DUAL ASCENT OPTIMIZATION

We solve the constrained problem equation 2-equation 3 using Lagrangian dual methods. The Lagrangian is:

$$\mathcal{L}(w, \phi, \lambda) = F(w) + \sum_{j \in \{E,C,M,T\}} \lambda_j \left( \frac{R_j(\phi)}{R_{j,\text{budget}}} - 1 \right) \tag{8}$$

where $\lambda = (\lambda_E, \lambda_C, \lambda_M, \lambda_T) \geq 0$ are dual variables and we normalize constraints by budget values for numerical stability.

The dual ascent updates are:

$$\lambda_j^{(t+1)} = \max\left(0, \lambda_j^{(t)} + \eta \cdot h\left(\frac{R_j(\phi^{(t)})}{R_{j,\text{budget}}}\right)\right) \tag{9}$$

where $h(x) = \max(0, x - 1.05)$ introduces a 5% dead-zone to prevent oscillations near constraint boundaries, and $\eta > 0$ is the dual learning rate.

[t] CAFL: Constraint-Aware Federated Learning [1] Initial model $w^{(0)}$, budgets $\{R_{j,\text{budget}}\}$, dual learning rate $\eta$ Trained model $w^{(T)}$ Initialize dual variables $\lambda^{(0)} = 0$ round $t = 0, 1, \ldots, T - 1$ $\phi^{(t)} \leftarrow \text{COMPUTEPOLICY}(\lambda^{(t)})$ Policy computation from dual variables

Sample clients $\mathcal{S}^{(t)} \subseteq [n]$ Client selection and local updates client $i \in \mathcal{S}^{(t)}$ $w_i^{(t+1)} \leftarrow \text{LOCALUPDATE}(w^{(t)}, \mathcal{D}_i, \phi^{(t)})$

$w^{(t+1)} \leftarrow \frac{1}{|\mathcal{S}^{(t)}|} \sum_{i \in \mathcal{S}^{(t)}} w_i^{(t+1)}$ Global aggregation

Measure resource usage $R_j(\phi^{(t)})$ for $j \in \{E, C, M, T\}$ Resource monitoring and dual updates Update dual variables: $\lambda_j^{(t+1)} = \max(0, \lambda_j^{(t)} + \eta \cdot h(R_j(\phi^{(t)})/R_{j,\text{budget}}))$

### 3.3 Adaptive Policy Design

The dual variables control an adaptive policy that modifies training parameters:

**Layer freezing depth:**

$$k_{\text{eff}} = \max\left(1, k_{\text{base}} - \lceil k_{\text{base}} \cdot \min(1, \xi_k p_{\text{depth}})\rceil\right) \tag{10}$$

where $p_{\text{depth}} = w_C \lambda_C + w_M \lambda_M + w_T \lambda_T$ combines constraint pressures, and $\xi_k, w_C, w_M, w_T$ are tunable coefficients.

**Training steps adaptation:**

$$s_{\text{eff}} = \max\left(s_{\min}, \lfloor s_{\text{base}} \cdot (1 - \min(\xi_s p_{\text{steps}}, 0.9))\rfloor\right) \tag{11}$$

where $p_{\text{steps}} = w_E \lambda_E + w_T' \lambda_T$ and $\xi_s, w_E, w_T'$ control step reduction.

**Batch size scaling:**

$$b_{\text{eff}} = \max\left(b_{\min}, \left\lfloor \frac{b_{\text{base}}}{1 + \xi_b p_{\text{batch}}} \right\rfloor\right) \tag{12}$$

where $p_{\text{batch}} = w_M' \lambda_M + w_T'' \lambda_T$ controls memory and thermal pressure response.

### 3.4 Token-Budget Preservation

To maintain training effectiveness under resource constraints, we preserve the total number of processed tokens per round through gradient accumulation. The target token count is:

$$N_{\text{target}} = s_{\text{base}} \cdot b_{\text{base}} \tag{13}$$

Given adapted parameters $(s_{\text{eff}}, b_{\text{eff}})$, we use gradient accumulation steps:

$$G = \left\lceil \frac{N_{\text{target}}}{s_{\text{eff}} \cdot b_{\text{eff}}} \right\rceil \tag{14}$$

This ensures consistent training signal while respecting physical resource constraints.

## 4 THEORETICAL ANALYSIS

### 4.1 CONVERGENCE OF DUAL ASCENT

We establish convergence properties of our dual ascent algorithm under standard assumptions.

**Assumption 1** (Resource Function Properties). The resource functions $R_j(\phi)$ are continuous and bounded for $\phi$ in the feasible parameter space $\Phi$.

**Assumption 2** (Policy Lipschitz Continuity). The policy function $\phi(\lambda)$ is Lipschitz continuous with constant $L_\phi$ in the dual variable space.

**Theorem 1** (Dual Variable Convergence). Under Assumptions 1 and 2, with dual learning rate $\eta \leq \frac{1}{2L_\phi \max_j \|R_j\|_\infty}$, the dual variables $\{\lambda^{(t)}\}$ converge to a neighborhood of the optimal dual solution $\lambda^*$ with:

$$\limsup_{t\to\infty} \|\lambda^{(t)} - \lambda^*\| \leq O(\epsilon_{\text{dead-zone}}) \tag{15}$$

where $\epsilon_{\text{dead-zone}} = 0.05$ is the dead-zone parameter.

**Proof Sketch.** The proof follows standard dual ascent convergence analysis. The dead-zone modification ensures that constraint violations within 5% do not generate dual updates, creating a bounded region around the optimal solution. The Lipschitz assumption on the policy function ensures that dual variable updates translate to bounded parameter changes, preventing instability.

### 4.2 RESOURCE CONSTRAINT SATISFACTION

**Theorem 2** (Constraint Satisfaction). Under the assumptions of Theorem 1, for sufficiently large $t$, the resource constraints are satisfied within the dead-zone tolerance:

$$\frac{R_j(\phi^{(t)})}{R_{j,\text{budget}}} \leq 1.05 + O(\epsilon_{\text{conv}}) \tag{16}$$

where $\epsilon_{\text{conv}} \to 0$ as $t \to \infty$.

### 4.3 TRAINING EFFECTIVENESS PRESERVATION

**Lemma 1** (Token-Budget Preservation). The gradient accumulation mechanism ensures that the expected number of processed tokens per client per round remains within $\epsilon_{\text{token}}$ of the baseline:

$$|\mathbb{E}[N_{\text{processed}}] - N_{\text{target}}| \leq \epsilon_{\text{token}} \tag{17}$$

where $\epsilon_{\text{token}}$ depends on the discretization error in computing accumulation steps.

## 5 EXPERIMENTAL EVALUATION

### 5.1 EXPERIMENTAL SETUP

**Dataset and Model.** We evaluate on character-level Shakespeare text modeling using a transformer architecture with 2 layers, 4 attention heads, and 128 embedding dimensions. The vocabulary size is 65 characters, and the sequence length is 128.

**Federated Configuration.** We simulate 16 clients with overlapping text shards (overlap of 48 characters). In each round, 6 clients are randomly selected for participation. Training runs for 50 rounds.

**Baselines.** We compare against standard FedAvg with fixed hyperparameters: 3 unfrozen layers, 60 local steps, batch size 16, and no compression.

**Implementation Details.** Dual learning rate $\eta = 0.01$, dead-zone parameter $\epsilon = 0.05$. Policy coefficients are tuned via grid search: $\xi_k = 1.8$, $\xi_s = 1.6$, $\xi_b = 8.0$.

### 5.2 MAIN RESULTS

Table 1 summarizes the key performance and resource usage metrics. CAFL achieves substantial resource savings while maintaining comparable loss convergence to the baseline (Figure 1).



Figure 1: Validation loss evolution showing both methods achieve similar initial convergence rates, with CAFL's constraints limiting further improvement in later rounds.

Table 1: Performance and resource usage comparison. CAFL achieves significant resource savings with acceptable accuracy trade-off.

| Method | Energy ($\times 10^6$) | Comm (MB) | Temp | Memory | Val. Loss |
|---|---|---|---|---|---|
| Budget Limit | 1.20 | 0.60 | 1.00 | 0.26 | – |
| FedAvg | 4.52 | 5.18 | 0.62 | 0.31 | 1.93 |
| CAFL-L | 1.35 | 0.28 | 0.57 | 0.24 | 2.10 |
| Improvement vs. FedAvg | 70%↓ | 95%↓ | 8%↓ | 23%↓ | 9%↑ |

## 5.3 Resource Constraint Analysis

Figures 2 and 3 demonstrate CAFL's effectiveness in managing multiple resource constraints simultaneously. CAFL successfully constrains energy usage and achieves dramatic communication savings compared to FedAvg, while maintaining memory usage within budget limits.

## 5.4 Trade-off Discussion

The loss gap represents a fundamental trade-off between model performance and resource efficiency. This degradation stems from three sources: **(1) Reduced model capacity:** Layer freezing limits the number of trainable parameters. **(2) Fewer training steps:** Energy constraints reduce local optimization iterations. **(3) Communication compression:** Quantization introduces gradient noise. However, this trade-off may be acceptable in many practical scenarios where resource constraints are hard limits.

## 6 Limitations and Future Work

**Resource modeling accuracy.** Our resource functions use simplified linear models. Real-world energy, thermal, and memory dynamics are more complex and device-dependent. Future work should incorporate device-specific profiling and nonlinear resource models.

**Heterogeneous client capabilities.** Current formulation assumes homogeneous resource budgets across clients. Extending to heterogeneous scenarios requires per-client constraint specification and adaptive client selection.

**Theoretical gaps.** While we establish dual convergence, we lack convergence rates for the federated learning objective under resource constraints. Future theoretical analysis should bound the impact of resource constraints on convergence speed and final accuracy.

**Alternative constraint optimization.** Beyond dual ascent, other constrained optimization approaches like augmented Lagrangian methods or barrier functions could be explored.

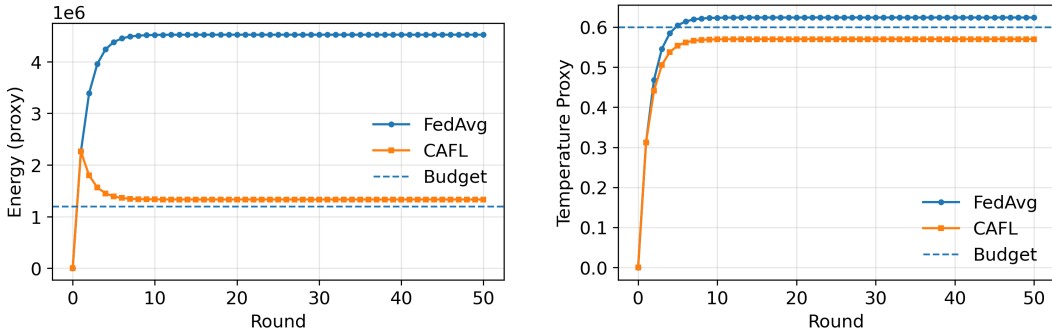

Figure 2: Energy and temperature control. CAFL-L prevents energy/thermal runaway by moderating computational intensity and staying near budget.

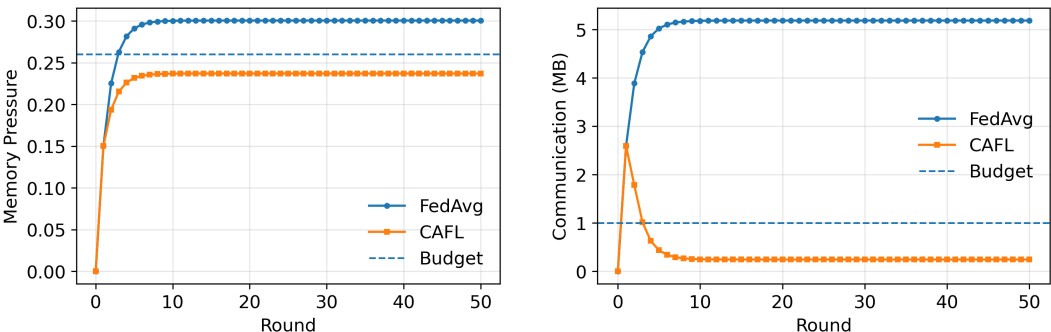

Figure 3: Resource-constraint satisfaction. CAFL-L adaptively manages memory and communication within budgets while FedAvg keeps violating them.

## 7 CONCLUSION

We introduced CAFL, a principled approach for multi-resource constrained federated learning that formulates the problem as constrained optimization and employs dual ascent for dynamic parameter adaptation. Our theoretical analysis establishes convergence properties of the dual controller, while experimental results demonstrate significant resource savings (70.5% energy, 95.3% communication) with reasonable accuracy trade-offs. CAFL represents a step toward practical federated learning deployment on resource-constrained edge devices. The framework's modularity enables extension to additional resource types and heterogeneous client scenarios, making it broadly applicable to real-world federated learning systems.

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
