# OpenReview forum: "Constraint-Aware Federated Learning: Multi-Resource Optimization via Dual Ascent"
_ICLR.cc/2026/Conference — Submitted to ICLR 2026_

### Official Review · Reviewer_utLZ · 2025-10-29

**Soundness:** 2
**Presentation:** 3
**Contribution:** 2
**Rating:** 2
**Confidence:** 5

**Summary:**

This paper presents CAFL (Constraint-Aware Federated Learning), which formulates federated learning as a constrained optimization problem considering multiple resource constraints (energy, communication, memory, and thermal). The authors employ Lagrangian dual ascent methods to dynamically adapt training parameters including layer freezing depth, batch size, and local training steps. Furthermore, the authors provide a theoretical convergence guarantee for the dual ascent controller. Their experiments on character-level Shakespeare text modeling demonstrate significant resource savings compared to FedAvg, notably 70.5% in energy and 95.3% in communication.

**Strengths:**

1）The paper tackles resource constraints, a crucial and highly practical challenge for the deployment of federated learning. It takes a systematic approach to federated learning under explicit, multi-dimensional resource constraints, providing a formal constrained optimization framework that is absent in much of the prior literature.

2）The paper is grounded in a mathematically well-motivated dual ascent method that provides clear convergence guarantees for the controller (as shown in Theorem 1 and 2) . Moreover, the introduction of adaptive policy mechanisms controlled by the dual variables shows a careful design that reflects the interplay between resource usage and learning dynamics.

3）The introduction of "token-budget preservation" is an interesting idea. By employing gradient accumulation to offset a reduced effective batch size or fewer local steps , the method intelligently decouples physical resource limitations from the algorithm's need to process a consistent number of samples per round.

**Weaknesses:**

1）The literature review is notably outdated and misses a significant body of recent, relevant research. The paper fails to position itself against modern state-of-the-art methods in both resource-constrained FL (e.g., FedRolex [1], DepthFL [2]) and advanced communication-efficient techniques [3, 4]. This lack of engagement makes it difficult to accurately assess the paper's novelty and contribution relative to the current landscape.

2）The models for energy and thermal management are overly simplistic. For instance, Equation 4's claim that energy consumption scales linearly with the number of trainable parameters requires much stronger theoretical or empirical justification, as energy use has complex dependencies on clock frequency, memory access patterns, and other non-trivial factors. Similarly, the thermal model (Eq. 7) is trivial and ignores key dynamics like heat dissipation and ambient temperature.

3）More critically, the framework models aggregate resource budgets for the entire round rather than on a per-client basis. This design choice fundamentally violates the premise of resource-constrained FL, where the bottleneck is the heterogeneous capability of individual devices. For example, if a single device's available memory is insufficient to afford local training, it cannot participate, a critical issue this aggregate model overlooks.

4）The experimental evaluation is not comprehensive enough to support the paper's broad claims. The only baseline used for comparison is FedAvg. The paper fails to benchmark against any other resource-aware, communication-efficient, or adaptive FL methods (as mentioned in point 1), making it impossible to judge its relative performance. The evaluation is restricted to a single, small-scale character-level language model (Shakespeare) using a 2-layer transformer. This is a traditional, simple task, and it is highly unclear if the proposed resource models, control mechanisms, and observed trade-offs would generalize to the larger, more modern architectures (e.g., Llama, ViT) and diverse modalities (e.g., vision) where resource constraints are most pressing.



[1] Fedrolex: Model-heterogeneous federated learning with rolling sub-model extraction, NeurIPS, 2022

[2] Depthfl: Depthwise federated learning for heterogeneous clients, ICLR, 2023

[3] Communication-efficient adaptive federated learning, ICML, 2022

[4] Feddm: Iterative distribution matching for communication-efficient federated learning, CVPR, 2023

**Questions:**

See Weaknesses.

---

### Official Review · Reviewer_jxqw · 2025-10-31

**Soundness:** 2
**Presentation:** 2
**Contribution:** 2
**Rating:** 4
**Confidence:** 5

**Summary:**

This paper proposes CAFL (Constraint-Aware Federated Learning), a framework that formulates federated learning (FL) as a multi-resource constrained optimization problem. The approach introduces a dual ascent controller that dynamically adjusts training parameters to satisfy multiple resource limits.

The authors also propose a token-budget preservation technique to maintain training progress under these adaptive constraints and provide theoretical convergence guarantees for the dual variables.

**Strengths:**

Addresses an important practical challenge in FL: real-world resource constraints across multiple modalities.
The dual ascent formulation is theoretically principled and integrates smoothly into the FL pipeline.
The paper is clearly written and logically structured.

**Weaknesses:**

Narrow experimental validation: Only one small-scale task with simulated clients. The setup lacks diversity and scale to demonstrate generality.

Limited baselines: Comparison only against FedAvg; misses other strong adaptive or efficiency-oriented FL algorithms.

Simplified resource modeling: Linear models for energy, thermal, and memory may not hold for real devices.

Theory–practice gap: Dual convergence is shown, but implications for federated convergence or model accuracy are not quantified.

Scalability and heterogeneity are untested, leaving unclear whether CAFL can operate effectively in realistic federated environments.

Weak contextualization: The paper doesn’t situate CAFL against key adaptive/resource-aware FL families (e.g., FedProx, FedAvgM/FedAdam, SCAFFOLD, compression/quantization, device-aware scheduling). A brief compare table (assumptions, constraints handled, objectives, comm/compute cost) would clarify what’s new.

Limited empirical breadth: Evaluation is confined to Shakespeare + one model, with no non-IID partitions, client/device heterogeneity (slow/fast, dropout), or scale variation. Add at least one more dataset/architecture and standard heterogeneity tests to support generality.

**Questions:**

Client heterogeneity:
How does CAFL handle heterogeneous clients with differing energy, memory, or communication budgets? Is the dual ascent mechanism still valid when resource limits vary per client, and can the dual variables be defined client-specifically without destabilizing training?

Baseline coverage:
Why were only FedAvg results reported? Methods such as FedProx, FedDyn, Scaffold, or FedAdapt target complementary aspects like system heterogeneity or communication efficiency. Even if these are not “unified,” why were they excluded as comparative baselines?

Resource-constrained methods:
How does CAFL compare against existing resource- or communication-efficient FL techniques (e.g., QSGD, Deep Gradient Compression, or energy-aware FL)? Without such comparisons, it is difficult to judge the effectiveness of the proposed framework relative to prior art.

Resource modeling assumptions:
The paper assumes linear resource functions for energy, temperature, and memory. What justifies this simplification, and how would the framework behave if more realistic nonlinear or empirically profiled models were used? Would convergence guarantees still hold?

Stability and sensitivity:
How sensitive is the algorithm to the choice of dual learning rate η and the dead-zone threshold? Were oscillations or instability observed when constraints were tight or when parameters were mis-specified?

Scalability and dataset scope:
Why was the evaluation limited to a single dataset and small-scale setup? Could CAFL be applied to larger or more standard FL benchmarks (e.g., CIFAR-10, FEMNIST, TinyImageNet) without significant modification? What prevents this extension in the current version?

---

### Official Review · Reviewer_z3TJ · 2025-11-02

**Soundness:** 2
**Presentation:** 2
**Contribution:** 1
**Rating:** 2
**Confidence:** 3

**Summary:**

The paper proposes CAFL, a dual-ascent controller that adapts layer freezing depth, local steps, batch size, and compression level to keep energy, communication, memory, and temperature within user-specified budgets during federated training. The method defines simple linear resource models (for energy/communication/memory/thermal), updates dual variables with a dead-zone, and maps them to training knobs via hand-tuned policy functions. Experiments on a simple Shakespeare with 16 clinets claim large energy/communication savings vs. FedAvg at modest performance degradation.

**Strengths:**

* Considering multiple constraints - energy, communication, memory, and thermal budgets - is important in FL.

**Weaknesses:**

* Baselines are oudated. The paper compares exclusively to FedAvg; it omits established baselines that already target some of these constraints (e.g., communication-efficient training/quantization/sparsification, system-heterogeneity methods, proximal/variance-reduction approaches).
* Resource modeling is overly simplistic and tuned. Energy, memory, and thermal equations are linear surrogates with global scaling constants and policy weights that are grid-searched; no device-specific profiling or validation against ground truth.
* Experimental setup is far from ICLR-level. Only a small, synthetic Shakespeare char-LM with 2 layers, 16 clients, and random participation; no realistic cross-device benchmarks (e.g., FedScale/LEAF, non-IID vision or mobile NLP), no on-device evaluation, no wall-clock/latency or user-experience metrics. Claims of energy/thermal benefits are therefore simulation-based, not measured.
* In addition, experimental results has no confidence intervals/error bars; no client-level fairness metrics; no heterogeneity scenarios (varying budgets or device classes); no ablations for η, dead-zone size, or policy weights.

**Questions:**

* Please see the weaknesses.

---

### Official Review · Reviewer_bdu6 · 2025-11-02

**Soundness:** 1
**Presentation:** 1
**Contribution:** 1
**Rating:** 0
**Confidence:** 3

**Summary:**

This paper formulates energy, communication, memory, and thermal constraints in a federated setting as a lagrangian equations, and attempts to limit the resource usage under the budget while conducting federated learning. The paper is only less than 6 pages, not well written, missing significant details on methodology and evaluation.

**Strengths:**

+ Lagrangian formulation of constraints in FL setting

**Weaknesses:**

The description of the dual ascent optimization seems incomplete. Line 118-128 seems truncated, missing punctuations, and could use rounds of polishing

There is a significant lack of details on how the Lagrangian formulation relates to the training process, and how that connects to resource usage reductions. The paper is only less than 6 pages long, and could use the 3-page spaces to fill those in.

1) Energy Consumption. Energy usage is proportional to computational workload. 2) Communication overhead depends on model size and compression. 3) Memory usage scales with batch size and model size. 4) Thermal load depends on computational intensity.

Clearly, the constraints are not orthogonal but correlated. It would be nice to rationalize the design to give each an independent lambda in the Lagrangian formulation. How does the covariance factor in?

How does the proposed method compare to existing related work? It would be helpful to compare with a wide spectrum of existing methods and conduct an ablation study.

**Questions:**

See weakness.

---

### Meta-Review · Area_Chair_czhf · 2025-12-31

**Summary:**

This paper proposes a Constraint-Aware Federated Learning (i.e., CAFL) method, which formulates federated learning as a constrained optimization problem considering multiple resource constraints including energy, communication, memory, and thermal. The CAFL uses Lagrangian dual methods to dynamically adapt training parameters including layer freezing depth, batch size, and local training steps. Moreover, it provides a convergence analysis for the proposed CAFL. It also provides some numerical experiments to verify efficiency of the proposed method.  However, its convergence analysis is very crude, and the results are unconvincing due to lacking detailed proofs. In the numerical experiments, many baselines (e.g., FedProx, FedDyn, Scaffold and FedAdapt ) for comparisons are missing.

The authors do not provide any rebuttal to  address reviewers' concerns. Since all reviewers suggest rejecting this paper, I agree with this assessment.

**Reviewer Concerns:**

The authors do not provide any rebuttal to the reviewers, so they do not address any reviewers' concerns.

**Reviewer Scores:**

Since the authors do not address any reviewers' concerns, the reviewers could not change their scores. All reviewers suggest rejecting this paper. I agree with this assessment.

---

### Decision · Program_Chairs · 2026-01-26

Reject